# Obstetric Violence from a Midwife Perspective

**DOI:** 10.3390/ijerph20064930

**Published:** 2023-03-10

**Authors:** Juan Miguel Martínez-Galiano, Julián Rodríguez-Almagro, Ana Rubio-Álvarez, Inmaculada Ortiz-Esquinas, Ana Ballesta-Castillejos, Antonio Hernández-Martínez

**Affiliations:** 1Nursing Department, University of Jaen, 23071 Jaen, Spain; 2Consortium for Biomedical Research in Epidemiology and Public Health (CIBERESP), 28029 Madrid, Spain; 3Department of Nursing, Physiotherapy and Occupational Therapy, Ciudad Real Faculty of Nursing, University of Castilla-La Mancha, 13071 Ciudad Real, Spain; 4Hospital Universitario de Torrejon, 28850 Madrid, Spain; 5Hospial Universitario Reina Sofia, 14004 Cordoba, Spain; 6Department of Nursing, Physiotherapy and Occupational Therapy, Albacete Faculty of Nursing, University of Castilla-La Mancha, 02008 Albacete, Spain

**Keywords:** obstetric violence, inadequate treatment, midwives, childbirth, human rights

## Abstract

This study examines and determines the prevalence of obstetric violence (OV) as perceived by midwives, as well as their knowledge of it and the professional factors that could be associated with the perception of OV. A cross-sectional study was conducted of 325 midwives in 2021 in Spain. Almost all (92.6%, 301) the midwives knew the term OV, but 74.8% (214) did not believe OV to be the same as malpractice. Moreover, 56.9% (185) stated they had rarely observed OV, and 26.5% (86) regularly observed OV. Most midwives consider physical aggression to be OV, in comparison, not providing information to women was only considered unacceptable treatment. The clinical practice considered the most grave within the context of OV was an instrumental birth or cesarean section without clinical justification. In addition, 97.5% (317) believed that raising awareness on the subject is one of the fundamental points to reducing this problem. Certain factors, such as less work experience, female gender, attendance at home births, and previous training in OV, were associated with an increased perception of situations as OV (*p* < 0.005). A high percentage of midwives perceived specific clinical practices (e.g., indicate cesarean section without clinical justification or perform the Kristeller maneuver) as OV, and certain characteristics of the professional profile, such as the professional experience or the sex of the midwife, were associated with an increased perception of OV. Most midwives knew the term OV but did not consider that it could pertain to some behaviors included in the international definitions of OV, such as the lack of information provided to a woman or the non-identification of the midwife, among others.

## 1. Introduction

Childbirth care has evolved from home-based care, where support and care were received from the family, to hospital-based care with hegemonic and technocratic medicine, where clinical practices are sometimes carried out that are not supported by scientific evidence [1].

In 2014, the World Health Organization (WHO) warned of the inappropriate, disrespectful, and even offensive treatment many women receive during birth assistance [2]. As a result, the concept of obstetric violence (OV) has become more visible. However, no clear and consensual international definition of OV exists, and validated instruments for detecting it are scarce [3]. A variety of behaviors that are considered OV can be found in the literature, which can be physical, verbal, or psycho-affective [4]. Common to all these behaviors is that they are behaviors, attitudes, or practices carried out by professionals during childbirth care and that, by action or omission, are perceived or may be perceived as violent or violating a woman’s human rights. In addition, there is also a common thread that includes OV as a form of gender violence [5,6].

The prevalence of women who have perceived inadequate treatment or OV during childbirth varies depending on the type of study and how OV is conceptualized. Despite this, the reported prevalence of OV in different studies is high, ranging from 25 to 78% [7,8,9,10,11,12,13]. In addition, a higher incidence of maternal and neonatal morbidity has been associated with a woman’s perception of experiencing OV during childbirth [14,15,16,17].

In general, health professionals are aware of the term OV but not the acts that may constitute it [18]. Some professionals perceive with discomfort that they are linked to OV, particularly as what is understood by OV they consider clinical practices belonging to their discipline, while others feel concerned about these practices being carried out [19]. Although health professionals report that women are treated with dignity and respect [20], some studies report that between 40 and 80% of health professionals have witnessed different situations that can be classified as OV [18,21,22]. In addition, 15–40% admit to having carried out some of these behaviors or practices [22,23]. Although other studies, such as one in Brazil with health professionals who attended deliveries, conclude that 70% of professionals state that they have never perpetrated OV [22]. Many healthcare professionals recognize the violence of certain practices but justify them based on the application of medical care [1]; however, others do not consider these practices violent [24].

In this sense, it seems that providing the necessary resources and training on the subject is one of the fundamental measures that professionals consider appropriate for improving this problem [25,26,27,28].

Currently, OV is a topic that generates social and professional controversy and sparks interest in our society [29], to the point that some countries have legislations to tackle it [30]. In addition, professional collectives have official positions regarding its existence and the use of the term OV [31,32,33,34,35].

In Spain, health professionals who attend the largest number of deliveries are midwives [36], Given the high number of women who report experiencing OV during childbirth, the scarcity of studies that address this issue, and even more so from the perspective of professionals, the disparity in the results, together with the strong social and professional demand we conducted this study to determine the percentage of OV and assess the knowledge and practices that may be related to it from the perception of midwives as well as explore the professional factors that could be associated with this perception.

## 2. Methods

### 2.1. Design and Participants

A cross-sectional observational study was conducted with midwives from September 2021 to December 2021. Retired midwives were excluded.

To estimate the required sample size, registered midwives in Spain, which amounted to 7639 midwives according to official statistics [37], were used as the reference population. A multi-response questionnaire was used, and the following parameters were used to estimate sample size: a prevalence of 50% as the most demanding criterion, a confidence level of 95%, precision or absolute error of 6%, and a replacement percentage of 10%. The resulting minimum sample size was 286 study subjects.

### 2.2. Data Collection and Data Sources

An online questionnaire was used comprising 54 items (2 open questions, 52 closed questions) that collected data on sociodemographic and professional characteristics (9), as well as questions related to the concept of obstetric violence, training on OV and its legal implications (10), various situations potentially compatible with OV (13), and questions regarding the perception of the degree of severity of various practices that have been related to OV in the previous literature (20).

The questionnaire was previously piloted, and a group of experts agreed on its content. It was distributed to midwives through the Federation of Midwife Associations of Spain (FAME, for its initials in Spanish) and the National Association of Midwives. The directors of these associations were involved in the dissemination of the project and the recruitment of participants. Before starting the questionnaire, the midwives were asked to read an information sheet about the study and its objectives. Then, they consented to participate in the study by checking the appropriate box in the questionnaire.

After agreeing to participate, they were given instructions to fill out the questionnaire. In addition, an email address was provided to respond to any questions or comments regarding the questionnaire.

### 2.3. Statical Analysis

The following variables were collected. The independent variables were: age, sex, works in a public center (No/Yes), works in a private center (No/Yes), attends home deliveries (No/Yes), works in primary care (No/Yes), annual number of births at work center (<1000 births, 1001–3000 births, >3000 births), if it is a teaching hospital “presence of professionals in training in your center” (No/Yes), experience as a midwife in birth (<5 years, 5–15 years, >15 years) and previous training in OV (No/Yes). The main dependent variable was the pooled perception of OV (No/Yes) in 13 situations that could arise in clinical practice. Originally, each of these clinical practices could be assessed as: it is neither OV nor inappropriate treatment; it is not OV, but is inappropriate treatment; it is not OV, but it is unacceptable treatment; it is minor OV; it is moderate OV; or it is grave OV. Other secondary dependent variables were the perception of severity when faced with certain practices, which in other studies have been previously related to OV.

First, a descriptive analysis was performed using absolute and relative frequencies for categorical variables and mean with standard deviation for numerical variables. Next, a multivariate analysis was performed by binary logistic regression using the SPSS ENTER method between the different sociodemographic and professional factors in relation to the pooled perception of OV in 13 clinical situations. The variables included in the model were: sex, previous training in obstetric violence, assists home births, experience as a midwife in the birth room, number of annual births at the workplace, teaching hospital, and works in a private center. The adjusted odds ratio (aOR) was estimated with its respective 95% confidence interval (95% CI).

### 2.4. Ethical Considerations

The Research Ethics Committee of the Integrated Care Management of Alcazar de San Juan Hospital approved this study, with ethical code 197-C. The ethical approval was approved at their meeting on 26 May 2021. Before starting the questionnaire, the midwives had to read an information sheet about the study and objectives and check a box in which they showed their consent to participate in it, that is, they signed a digital informed consent prepared ad hoc.

All methods were carried out in accordance with relevant guidelines and regulations. Informed consent was obtained from all subjects.

## 3. Results

### Characteristics of Participants

A total of 325 midwives participated, of which 91.7% (298) were women. Other sample characteristics of note were: 8.9% (29) of the sample attending home births and 31.7% (103) of the midwives had participated in a workshop and/or course on OV. Table 1 presents the professional characteristics and the work environment in detail.

Regarding the term OV, 92.6% (301) were aware of the term, and 37.8% (123) completely agreed with the definition established by the WHO, while 9.5% (31) completely disagreed. When asked if the term OV was the same as professional malpractice, 74.2% (241) answered that it was not the same. Around 44.0% (143) of midwives considered that this problem needed to be addressed but was not being approached correctly in Spain, and 15.1% (49) felt that professionals were being excessively criminalized. Regarding the frequency of this problem, 56.9% (185) affirmed that they had observed OV, but only rarely, while 26.5% (86) stated that the OV occurred regularly in their unit.

Regarding the legal implications, 79.4% (258) knew that a specific legislation was being worked on, and 44.9% (146) considered that this law would improve their clinical practice. Along the same lines, different measures were proposed to the midwives to eradicate OV. Although all suggestions were strongly supported, the most supported measure was “Greater awareness of professionals” with 97.5% (317), and the least supported was “A tougher legislation for the professionals who carry it out” with 61.8% (201). The proposals and support are detailed in Table 2.

Next, we presented 13 clinical situations that could occur in routine clinical practice to determine the current perception of each scenario of the midwives included in the present study. Each of these situations could be assessed as: it is neither OV nor inappropriate treatment; it is not OV, but is inappropriate treatment; it is not OV, but is unacceptable treatment; it is minor OV; it is moderate OV; or it is grave OV. Of these 13 situations, nine presented the highest percentage of responses considering them to be grave OV, with percentages of responses that ranged between 32.6% (106) for “Treating the women as a child” and 85.8% (279) for “Physically assaulting the women”. On the other hand, 40.6% (132) considered that “Delaying care without justification” is not OV, instead is inappropriate treatment. The highest number of responses for “Lack of identification” and “Insufficient reporting” considered that these situations were not OV but unacceptable treatment, with 28.0% (91) and 27.7% (90), respectively (Table 3).

Next, we aimed to determine which sociodemographic and professional factors could be related to perceiving these situations as OV. For this, the previous results were grouped in a dichotomous variable “grouped perception of obstetric violence”. As shown in Table 4, four variables were related to an increased perception of situations as OV. The variable repeated the most is the time worked in the birth room. For 11 of the 13 situations, midwives with 5–15 years of work experience and more than 15 years perceived these situations as OV less often compared to those who had worked in the birth room for less than 5 years. The sex of the midwife was also associated with the perception of situations as OV; specifically, being a woman was associated with an increased perception of situations as OV in six of the thirteen situations. The third most frequent variable was previous OV training, which was related to increased perception of OV in four situations, and assistance in home births in three situations. That is, those midwives who had previous training in OV and who attended home deliveries more frequently perceive the existence of OV in the situations described.

Finally, the midwives were presented with 20 clinical practices that have been associated with OV in other studies to determine the degree of severity of each one according to their perception. The highest number of responses for all 20 practices considered each as very grave OV; hence, the average score was used to rank the practices for the degree of OV severity. The practice with the highest score for severity was a “cesarean section without clinical justification”, followed by “the performance of an instrumental birth without clinical justification”, and “the administration of sedative drugs without justification or consent”. All detailed responses are presented in Table 5.

## 4. Discussion

The present study aimed to determine the prevalence of OV as perceived by midwives and assess their knowledge of OV. Our findings indicate that most midwives are familiar with the term OV but do not identify it as a synonym for professional malpractice. In addition, most stated that they have rarely witnessed OV, while a quarter reported that it is common in their workplace. Among the possible behaviors that can be considered OV, the majority of the midwives recognized physical aggression as one of them. However, most did not consider that insufficient reporting or a lack of introduction to the women could be considered OV, although they did classify these as unacceptable practices. Similarly, delaying care without justification was not considered OV but inappropriate treatment. Among the clinical practices considered OV, a cesarean section or instrumental birth without clinical justification stands out with the highest score. Considering that these practices are influenced by defensive medicine [38], this high score may be due to differing perspectives between the midwives in this study and other health professionals providing obstetric care regarding defensive medicine, with the midwives not justifying a high number of cesareans and considering it OV. Regarding professional profile, female midwives, those with less professional experience, those with previous OV training, and those who attend births at home were more likely to perceive specific situations as OV.

Coinciding with the results of Faneite et al. who carried out a study in Venezuela with 500 obstetric professionals from different centers [18], midwives knew the term OV and agreed with it, despite the existing controversies about its concept [32]. This may be due to the essential role that different digital social platforms and communication media are playing, where women have been able to disseminate their experiences and spread awareness of the problem. In addition, the positioning of different professional associations with respect to the subject and the dissemination they have been able to make among their associates has also contributed to the knowledge of the term OV, along with the information provided in WHO reports and scientific articles on the subject of OV [31,32,33].

Most midwives did not identify OV solely as professional malpractice, as many different groups (e.g., associations of health professionals related to birth assistance) try to assert. Recently in Spain, due to the visibility that this form of violence is gaining and the concerns about it, legislative processes have been initiated to categorize it and penalize professionals who, by action or omission, carry out harmful procedures against women, as some Latin American countries have previously conducted (30). Our results show that 44% of midwives believe it is necessary to address the problem, although they consider it not being executed correctly. Legislation in this regard may help improve clinical practice. The midwives proposed solutions aimed at better training, information, and awareness in line with what is proposed in different studies carried out in Brazil [39,40,41,42], Ethiopia [25,26], and an integrative review of the literature carried out by Barbosa Jardim and Modena [43].

In line with what was found by different authors [18,21,22], most of the midwives (92%) reported situations of OV in their workplace. This prevalence is 12 percentage points above the highest prevalence detected in the literature, which Costa Cardoso et al. found in a study carried out in Brazil, where 80% of the professionals had witnessed OV behaviors carried out by their peers. Yet, in the same study, 70% of the professionals denied having carried out OV [22], and instead reported having treated women with dignity and respect—a professional perception similar to that detected by A Afulani et al. [20]. This may be because various clinical practices that can be considered OV are standardized as typical of clinical practice (1,24,41). Hence, there is a need for training as an element on which to base the prevention and elimination of OV, as identified in our results and other studies [25,41,43].

In identifying the behaviors that could be classified as OV, midwives mostly considered those related to aggression in line with results from Burrowes et al. [26] and others [39,44]. Other forms more subtle of OV, such as delaying health care without justification, healthcare professionals not identifying themselves, inadequate reporting, or not informing the woman’s main support person were not considered OV by midwives but were considered inappropriate or unacceptable treatment, which are results that contrast with those found by Teles de Alexandria et al. [39] in which they did classify said behaviors as OV. All clinical practices not supported by scientific evidence and that which could be carried out in birth rooms are considered by most midwives to be very grave OV, in line with what several authors reported [39,44]. Although Pito Leal et al. found that some professionals did not consider some of these practices as OV [45], there exists findings that are in agreement with other authors [18,24,46]. The midwives considered the gravest practice regarding OV to be in the birth room performing a cesarean section or an instrumental birth without clinical justification. These results coincide with those of Prado Murrieta in a study carried out in Mexico with obstetric and gynecological physicians and hospital workers [47].

Midwives who were women, as Mayra et al. [48] also identified, with previous OV training, less work experience attending births, and those who attended births at home were those who identified a greater number of behaviors that are considered OV. This may be because, over time and with increasing experience, inadequate behaviors are normalized and integrated as habitual and, as a result, are not classified as OV as they are considered typical of the discipline; a conclusion also made in other studies carried out in Uruguay [1] and Ecuador [24]. Once again, training is essential, as this promotes more information and awareness to recognize the different behaviors and practices that OV encompasses so that they are avoided and can be eradicated from birth rooms, a solution also proposed by different authors [41,43]. Of note, the type of hospital (regardless of the number of women who attended during delivery or whether it is a teaching hospital) where the midwives worked was not associated with the perception of the existence of OV, in line with what Moyer et al. [49] found.

Regarding the limitations of this study, as it is a questionnaire, a selection bias associated with non-response is possible; however, the sample is large, and we consider that it is representative, so we do not believe that the responses of the midwives who did not participate would differ greatly from those who did. Furthermore, the number of midwives who refused to participate was low, at 21 midwives. The existence of an information bias is unlikely due to the characteristics of the data collected, and the possible answers provided were simple, straightforward, and understandable. A recall bias cannot be completely ruled out, although the information was collected on clinical practice that the midwife experiences on a daily basis in her workplace, so if the influence on the results had been produced, we believe that it would be minimal. The midwives’ responses may be oriented toward what is socially acceptable in such a way that the perception of OV is denied, but the percentage of midwives who claim to have perceived OV situations is high and with figures similar to and higher than the few other published studies. Hence, we believe this response bias has not occurred, and they responded without that limitation. Nor is it possible to completely rule out a confounding bias despite attempting to control both through the study design and data analysis, thereby adjusting for variables that could influence the results. One of the study’s main limitations is the lack of unification regarding the concept and behaviors that OV encompasses.

The questionnaire was presented online, which perhaps limited the participation of midwives who do not have internet access, although this is rare as most of the population has devices (smartphones, tablets, computers, etc.) with internet connection. An online questionnaire has already been used previously as an instrument for data collection in various investigations.

## 5. Conclusions

In conclusion, the percentage of midwives who admit that they have witnessed OV is high. Overall, midwives know the term OV and agree with it. However, some of the behaviors that OV encompasses, such as a professional not identifying themselves to a woman in their care or not providing enough information, are not considered OV by midwives. All the clinical practices that can be carried out in a birth room and that are not supported by scientific evidence, such as the Kristeller maneuver and the performance of an episiotomy without justification, among others, are recognized by midwives as very grave OV, highlighting that performing a cesarean section or instrumental birth without clinical justification as one of the gravest acts of OV. OV is an indicator of poor quality of care with short- and long-term maternal repercussions. Therefore, the training of professionals, the standardization of care through the application of clinical practice guidelines, and health initiatives that encourage the humanization of childbirth will be key to the eradication of OV. A high percentage of midwives perceived specific clinical practices as OV, and the characteristics of the professional profile are associated with the likelihood of perceiving certain situations as OV.

## Figures and Tables

**Table 1 ijerph-20-04930-t001:** Sample characteristics.

Variable	n (%)
**Age**	
≤25 years	12 (3.7)
26–35 years	117 (36.0)
36–45 years	97 (29.8)
46–55 years	60 (18.5)
56–65 years	39 (12.0)
**Sex**	
Male	27 (8.3)
Female	298 (91.7)
**Duration of work experience in the birth room**	
<5 years	109 (33.5)
5–15 years	130 (40.0)
>15 years	86 (26.5)
**Works in a public center**	
No	10 (3.1)
Yes	315 (96.9)
**Works in a private center**	
No	269 (82.8)
Yes	56 (17.2)
**Provides home-birth care**	
No	296 (91.1)
Yes	29 (8.9)
**Yearly hospital deliveries**	
<1000 deliveries	102 (31.4)
1000–3000 deliveries	166 (51.1)
>4000 deliveries	57 (17.5)
**Teaching hospital (midwife and obstetric training)**	
No	66 (20.3)
Yes	259 (79.7)
**Previous training in obstetric violence (attended a workshop and/or seminary/course)**	
No	222 (68.3)
Yes	103 (31.7)

**Table 2 ijerph-20-04930-t002:** Questions regarding the perception of obstetric violence and its implications.

Questions for All Professionals	n (%)
**Do you know what obstetric violence is?**	
No	5 (1.5)
Yes	301 (92.6)
I’m not very clear on what it entails	19 (5.8)
**According to the WHO “Obstetric violence is that what women suffer during pregnancy or childbirth when receiving physical abuse, verbal abuse or humiliation, or coercive or unconsented medical procedures”. Do you agree with this definition?**	
Completely disagree	31 (9.5)
Somewhat disagree	24 (7.4)
Neither agree nor disagree	18 (5.5)
Somewhat agree	129 (39.7)
Completely agree	123 (37.8)
**Do you consider obstetric violence to be the same as professional malpractice?**	
No	241 (74.2)
Yes	84 (25.8)
**In recent months obstetric violence has been discussed a lot in different** **communication media. How does it affect you and/or how you feel as a professional?**	
I feel that they are finally addressing this very important problem	130 (40.0)
I feel that it is necessary to address this problem, but it is not being addressed in the right way	143 (44.0)
I feel that it is excessively criminalizing the professionals	49 (15.1)
I am indifferent	3 (0.9)
**In your professional development have you noted conducts that could be considered obstetric violence?**	
Obstetric violence does not exist, I do not agree with the term	16 (4.9)
I have never witnessed any obstetric violence	10 (3.1)
I have only rarely witnessed obstetric violence	185 (56.9)
Obstetric violence is habitual in my unit	86 (26.5)
Obstetric violence is very frequent in my unit	28 (8.6)
**Were you aware that obstetric violence is considered a crime in some countries?**	
No	147 (45.2)
Yes	178 (54.8)
**Were you aware that Spain intends to legislate on obstetric violence?**	
No	67 (20.6)
Yes	258 (79.4)
**How would specific legislation influence your clinical practice?**	
It would not change it	107 (32.9)
It would influence it toward more defensive medicine or clinical practice	72 (22.2)
It would influence it by improving it	146 (44.9)
**Do you recognize yourself in the commentaries regarding obstetric violence in the media?**	
No	260 (80.0)
Yes	65 (20.0)
**What measures would you introduce to try to eradicate OV?**	
**More formation for healthcare professionals**	
Yes	290 (89.2)
No	9 (2.8)
I’m not clear	26 (8.0)
**Raise awareness among healthcare professionals**	
Yes	317 (97.5)
No	3 (0.9)
I’m not clear	5 (1.5)
**More realistic information for women and their families**	
Yes	300 (92.3)
No	6 (1.8)
I’m not very clear	19 (5.8)
**Increased involvement of women in the childbirth process**	
Yes	301 (92.6)
No	4 (1.2)
I’m not very clear	20 (6.2)
**More strict legislations for the professionals involved**	
Yes	201 (61.8)
No	37 (11.4)
I’m not clear	87 (26.8)
**More protocols to avoid variability**	
Yes	248 (76.3)
No	18 (5.5)
I’m not clear	59 (18.2)
**External audits**	
Yes	274 (84.3)
No	14 (4.3)
I’m not clear	37 (11.4)

**Table 3 ijerph-20-04930-t003:** Perception of the study midwives of specific situations in clinical practice.

Clinical Situations	How Do you Perceive This Situation?					
	Is Not Obstetric Violence nor Inadequate Treatment	Is Not Obstetric Violence, but It Is Inadequate Treatment	Is Not Obstetric Violence, but It Is Unacceptable Treatment	Is Minor Obstetric Violence	Is Moderate Obstetric Violence	Is Grave Obstetric Violence
Delay treatment without justification	5 (1.5)	**132 (40.6)**	105 (32.3)	31 (9.5)	31 (9.5)	21 (6.5)
Provide insufficient information	6 (1.8)	83 (25.5)	**90 (27.7)**	46 (14.2)	57 (17.5)	43 (13.2)
Perform a procedure without consent	7 (2.2)	26 (8.0)	52 (16.0)	23 (7.1)	33 (10.2)	**184 (56.6)**
Reprimand a woman for their behavior	11 (3.4)	29 (8.9)	45 (13.8)	26 (8.0)	49 (15.1)	**165 (50.8)**
Treat the women with contempt	8 (2.5)	13 (4.0)	45 (13.8)	11 (3.4)	27 (8.3)	**221 (68.0)**
Criticize the women for their behavior	10 (3.1)	23 (7.1)	53 (16.3)	17 (5.2)	67 (20.6)	**155 (47.7)**
Threaten the women for their behavior	7 (2.2)	16 (4.9)	31 (9.5)	8 (2.5)	22 (6.8)	**241 (74.2)**
Physically assault the women (slap, hit legs, etc.)	9 (2.8)	10 (3.1)	18 (5.5)	1 (0.3)	8 (2.5)	**279 (85.8)**
Perform a practice that is not supported by scientific evidence	9 (2.8)	28 (8.6)	61 (18.8)	15 (4.6)	66 (20.3)	**146 (44.9)**
Not ensure the privacy of the women	5 (1.5)	36 (11.1)	50 (15.4)	25 (7.7)	62 (19.1)	**147 (45.2)**
Not introduce yourself with your name and/or position the first time when caring for a woman when it is not an emergency	9 (2.8)	79 (24.3)	**91 (28.0)**	33 (10.2)	69 (21.2)	44 (13.5)
Treat the women as a children	7 (2.2)	48 (14.8)	64 (19.7)	34 (10.5)	66 (20.3)	**106 (32.6)**
Not provide information to the woman’s main support person	15 (4.6)	76 (23.4)	**91 (28.0)**	34 (10.5)	63 (19.4)	46 (14.2)

The answer with the highest number of responses is indicated in bold.

**Table 4 ijerph-20-04930-t004:** Factors related to the perception of obstetric violence among midwives.

Clinical Situations	aOR 95% CI of the Different Sociodemographic and Professional Factors for the Perception of Obstetric Violence on Midwives						
	Female Sex (RC: Male)	Previous Training on Obstetric Violence (RC: No)	Attends Home Deliveries (RC: No)	Duration Working in Birth Rooms (RC: <5 Years)	Hospital Size by Number of Births (RC: <1000 Deliveries)	Teaching Hospital (RC: No)	Private Center (RC: No)
Delay treatment without justification	1.07 (0.40–2.89)	**2.71 (1.57–4.67)**	1.86 (0.80–4.37)	**5–15 years: 0.46 (0.25–0.84)** **>15 years: 0.48 (0.24–0.95)**	1000–3000: 1.12 (0.53–2.39) >3000; 1.14 (0.47–2.84)	0.71 (0.31–1.63)	1.09 (0.55–2.15)
Provide insufficient information	1.80 (0.73–4.46)	**2.07 (1.25–3.45)**	2.15 (0.90–5.10)	**5–15 years: 0.42 (0.25–0.73)** **>15 years: 0.32 (0.18–0.60)**	1000–3000: 1.36 (0.71–2.60) >3000; 0.82 (0.37–1.85)	1.09 (0.52–2.28)	0.54 (0.28–1.04)
Perform a procedure without consent	1.85 (0.79–4.30)	1.58 (0.87–2.84)	**4.66 (1.05–20.72)**	**5–15 years: 0.47 (0.25–0.90)** **>15 years: 0.39 (0.19–0.78)**	1000–3000: 1.16 (0.57–2.34) >3000; 1.54 (0.62–3.81)	0.80 (0.36–1.77)	0.82 (0.41–1.64)
Reprimand a woman for their behavior	**3.19 (1.38–7.34)**	1.75 (0.96–3.19)	2.75 (0.77–9.82)	5–15 years: 0.58 (0.31–1.11) **>15 years: 0.40 (0.20–0.79)**	1000–3000: 1.06 (0.52–2.17) >3000; 1.09 (0.45–2.66)	1.05 (0.47–2.36)	1.10 (0.53–2.25)
Treat the women with contempt	**3.30 (1.42–7.67)**	1.53 (0.80–2.92)	2.07 (0.58–7.43)	5–15 years: 1.18 (0.59–2.36) >15 years: 0.58 (0.29–1.17)	1000–3000: 1.15 (0.54–2.45) >3000; 1.28 (0.50–3.31)	0.63 (0.26–1.54)	0.90 (0.42–1.92)
Criticize the women for their behavior	**3.69 (1.60–8.49)**	1.70 (0.94–3.07)	2.07 (0.66–6.46)	5–15 years: 0.64 (0.34–1.22) >15 years: 0.40 (0.20–0.79)	1000–3000: 1.05 (0.51–2.17) >3000; 0.88 (0.37–2.13)	0.97 (0.43–2.18)	0.85 (0.43–1.71)
Threaten the women for their behavior	**3.93 (1.64–9.39)**	1.49 (0.74–3.01)	2.61 (0.57–11.98)	5–15 years: 0.85 (0.39–1.84) **>15 years: 0.43 (0.20–0.93)**	1000–3000: 0.61 (0.26–1.44) >3000; 1.38 (0.43–4.44)	1.00 (0.38–2.66)	0.95 (0.42–2.17)
Physically assault the women (slap, hit legs, etc.)	2.38 (0.88–6.47)	1.04 (0.48–2.25)	1.72 (0.38–7.90)	5–15 years: 0.82 (0.35–1.94) >15 years: 0.66 (0.27–1.65)	1000–3000: 0.73 (0.28–1.93) >3000; 1.37 (0.36–5.26)	1.36 (0.47–3.93)	1.09 (0.55–2.15)
Perform a practice that is not supported by scientific evidence	**2.71 (1.19–6.20)**	1.27 (0.74–2.19)	1.66 (0.61–4.32)	**5–15 years: 0.42 (0.23–0.77)** **>15 years: 0.50 (0.25–0.97)**	1000–3000: 0.70 (0.34–1.41) >3000; 0.98 (0.40–2.38)	1.32 (0.60–2.89)	0.78 (0.41–1.49)
Not ensure the privacy of the women	**2.37 (1.02–5.50)**	**1.86 (1.03–3.37)**	3.28 (0.93–11.63)	5–15 years: 0.56 (0.29–1.07) **>15 years: 0.33 (0.17–0.65)**	1000–3000: 1.10 (0.54–2.24) >3000; 0.70 (0.30–1.64)	0.71 (0.31–1.62)	0.97 (0.48–1.96)
Not introduce oneself with name or position the first time caring for a woman when it is not an emergency	2.20 (0.88–5.55)	**1.79 (1.09–2.94)**	1.69 (0.73–3.93)	**5–15 years: 0.55 (0.32–0.93)** **>15 years: 0.39 (0.21–0.71)**	1000–3000: 1.36 (0.71–2.60) >3000; 1.21 (0.54–2.67)	0.93 (0.45–1.93)	0.94 (0.51–1.74)
Treat the women as a child	2.07 (0.88–4.85)	1.62 (0.94–2.78)	**5.39 (1.53–19.03)**	**5–15 years: 0.33 (0.18–0.60** **>15 years: 0.25 (0.13–0.47)**	1000–3000: 1.40 (0.71–2.76) >3000; 0.79 (0.34–1.79)	0.73 (0.34–1.56)	0.82 (0.43–1.59)
Not provide information to the woman’s main support person	2.61 (0.99–6.87)	1.56 (0.94–2.58)	2.34 (0.98–5.61)	**5–15 years: 0.40 (0.23–0.68)** **>15 years: 0.37 (0.20–0.68)**	1000–3000: 1.70 (0.86–3.33) >3000; 1.47 (0.64–3.38)	0.49 (0.23–1.03)	1.02 (0.27–3.90)

aOR: Adjusted odds ratio; RC: Reference category; Bold: Statistically significant differences.

**Table 5 ijerph-20-04930-t005:** Assessment by midwives of the degree of OV severity of different clinical practices not recommended by scientific evidence.

Clinical Practices without Evidence or Not Recommended	How Do You Perceive This Practice?						
	Not at All Grave >1 POINTS	Minor >2 POINTS	Somewhat Grave >3 POINTS	Quite Grave >4 POINTS	Very Grave >5 POINTS	Average Score	Order of Severity *
Separation of mother–infant without medical justification	1 (0.3)	9 (2.8)	21 (6.5)	59 (18.2)	**235 (72.3)**	4.59 (0.76)	4
Perform an episiotomy without justification	1 (0.3)	5 (1.5)	27 (8.3)	65 (20.0)	**227 (69.8)**	4.58 (0.74)	6
Administer an enema without consent	2 (0.6)	11 (3.4)	39 (12.0)	81 (24.9)	**192 (59.1)**	4.38 (0.87)	9
Administer oxytocin to speed up labor without clinical justification	4 (1.2)	15 (4.6)	47 (14.5)	85 (26.2)	**174 (53.5)**	4.26 (0.95)	14
Indicate cesarean section without clinical justification	1 (0.3)	4 (1.2)	13 (4.0)	20 (6.2)	**287 (88.3)**	4.81 (0.59)	1
Perform an instrumental birth without clinical justification	1 (0.3)	2 (0.6)	16 (4.9)	24 (7.4)	**282 (86.8)**	4.80 (0.58)	2
Perform the Kristeller maneuver	7 (2.2)	30 (9.2)	46 (14.2)	75 (23.1)	**167 (51.4)**	4.12 (1.10)	17
Perform amniorrhexis without clinical justification	4 (1.2)	23 (7.1)	57 (17.5)	89 (27.4)	**152 (46.8)**	4.11 (1.01)	18
Not permit the woman to mobilize without clinical justification	1 (0.3)	13 (4.0)	48 (14.8)	79 (24.3)	**184 (56.6)**	4.33 (0.89)	11
Prohibit the intake of fluids without clinical justification	1 (0.3)	16 (4.9)	58 (17.8)	93 (28.6)	**157 (48.3)**	4.20 (0.92)	15
Administer sedatives without justification or consent	2 (0.6)	3 (0.9)	20 (6.2)	30 (9.2)	**270 (83.1)**	4.73 (0.67)	3
Shorten the expulsive phase with different procedures without clinical justification	2 (0.6)	11 (3.4)	27 (8.3)	68 (20.9)	**217 (66.8)**	4.51 (0.83)	7
Coerce a woman to adopt a specific position during labor	2 (0.6)	13 (4.0)	40 (12.3)	92 (28.3)	**178 (54.8)**	4.33 (0.88)	12
Interrupt and/or reduce the epidural analgesia dose without consent	4 (1.2)	16 (4.9)	43 (13.2)	79 (24.3)	**183 (56.3)**	4.30 (0.96)	13
Perform internal monitoring without clinical justification	5 (1.5)	12 (3.7)	37 (11.4)	78 (24.0)	**193 (59.4)**	4.36 (0.93)	10
Remove the partner from the room to perform certain interventions (amniorrhexis, exploration, monitoring, instrumental birth, etc.)	3 (0.9)	19 (5.8)	53 (16.3)	92 (28.3)	**158 (48.6)**	4.18 (0.97)	16
Initiate directed pushes without clinical justification	7 (2.2)	30 (9.2)	66 (20.3)	106 (32.6)	**116 (35.7)**	3.90 (1.06)	20
Not respect the culture or rituals of the woman as long as they do not interfere with scientific evidence	9 (2.8)	24 (7.4)	54 (16.6)	83 (25.5)	**155 (47.7)**	4.08 (1.09)	19
Carry out practices that put breastfeeding at risk if the woman wishes to breastfeed	1 (0.3)	12 (3.7)	31 (9.5)	60 (18.5)	**221 (68.0)**	4.50 (0.84)	8
Perform repeated internal vaginal examinations by different professionals without consent	3 (0.9)	7 (2.2)	28 (8.6)	45 (13.8)	**242 (74.5)**	4.59 (0.81)	5

The answer with the highest number of responses is indicated in bold; * Ranking of the clinical practices in terms of perceived severity of OV. The ranking is based on the average score.

## Data Availability

The data that support the findings of this study are available from the corresponding author, upon reasonable request.

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
