# Peer review of "Obstetric Violence from a Midwife Perspective"

_ijerph, 2023, doi:10.3390/ijerph20064930_

Round 1

Reviewer 1 Report

Thank you for the opportunity to read the manuscript. The paper raises a very important issue of obstetric violence. The study seems to have been prepared in the right way. The results are presented clearly. The discussion is also conducted correctly, and the conclusions drawn are justified. 

Comments: please remove the double numbering in the references

Author Response

RESPONSE TO REVIEWER1

2 March 2023

Dear Editor of International Journal of Environmental Research and Public Health,

Thank you very much for the opportunity to revise and improve the Manuscript ID: ijerph-2175134 entitled “Violence from the perspective of midwives”

First, we want to express our sincere thanks to the reviewers, as well as their constructive comments. Their comments have allowed us not only to significantly improve the manuscript, but also to reflect on future research in the field of in women' health.

Following this letter, we detail point-by-point our comments in response to the suggestions of the reviewers and the changes that we have made in the revised version of our manuscript. In the revised manuscript, we have highlighted the modifications made to the original text. We hope that the improvements made along with the associated responses below, deserve the definitive approval of the Editorial Team International Journal of Environmental Research and Public Health. If not, all authors remain at your disposal to resolve any issue.

We look forward to hearing from you at your earliest convenience.

Sincerely,

Juan Miguel Martínez Galiano

Reviewer 1 : Comments and Suggestions for Authors:

Thank you for the opportunity to read the manuscript. The paper raises a very important issue of obstetric violence. The study seems to have been prepared in the right way. The results are presented clearly. The discussion is also conducted correctly, and the conclusions drawn are justified. 

Comments: please remove the double numbering in the references

Author´s Response: Thank you very much for your comments. We have eliminated double references

Reviewer 2 Report

After examining the scientific study, the following considerations may be made. The scientific study is well structured in all its parts. In particular, the premises with which the authors introduced the analysis are clear. Equally clear are the objectives that led the authors to carry out this study and the section on materials and methods. Particular appreciation can also be expressed for the material on which the study was carried out. The data was collected methodically and without bias. The results were consistent and significant and allowed a discussion section full of food for thought. The authors then developed a discussion of the results achieved.

The number and quality of the citations are appropriate; however, the scientific relevance of the article could benefit from an expansion of the same. In the specific advice to add the following quotes:

In the discussion section I would suggest to underlined that the decision to choose a cesarean section rate is associated with the phenomenon of defensive medicine. Thus I suggest adding the following quote: " Fineschi V, Arcangeli M, Di Fazio N, Del Fante Z, Fineschi B, Santoro P, Frati P, Associazione Consulcesi Health And Onlus Futura Ricerca. Defensive Medicine in the Management of Cesarean Delivery: A Survey among Italian Physicians. Healthcare (Basel). 2021 Aug 25;9(9):1097. doi: 10.3390/healthcare9091097. PMID: 34574870; PMCID: PMC8472348.”

English is well structured in syntax and grammar.

Author Response

RESPONSE TO REVIEWER2

2 March 2023

Dear Editor of International Journal of Environmental Research and Public Health,

Thank you very much for the opportunity to revise and improve the Manuscript ID: ijerph-2175134 entitled “Violence from the perspective of midwives”

First, we want to express our sincere thanks to the reviewers, as well as their constructive comments. Their comments have allowed us not only to significantly improve the manuscript, but also to reflect on future research in the field of in women' health.

Following this letter, we detail point-by-point our comments in response to the suggestions of the reviewers and the changes that we have made in the revised version of our manuscript. In the revised manuscript, we have highlighted the modifications made to the original text. We hope that the improvements made along with the associated responses below, deserve the definitive approval of the Editorial Team International Journal of Environmental Research and Public Health. If not, all authors remain at your disposal to resolve any issue.

We look forward to hearing from you at your earliest convenience.

Sincerely,

Juan Miguel Martínez Galiano

Reviewer 2: Comments and Suggestions for Authors:

(x) English language and style are fine/minor spell check required

Author´s Response: Than you very much for your comments. Dr. Ingrid de Ruiter is a Medical Writer and Public Health Researcher with a background in Clinical Medicine and Public Health. Professional Member of European Medical Writers Association and American Medical Writers Association. She has carefully checked the manuscript. We think no more grammatical errors appear.

After examining the scientific study, the following considerations may be made. The scientific study is well structured in all its parts. In particular, the premises with which the authors introduced the analysis are clear. Equally clear are the objectives that led the authors to carry out this study and the section on materials and methods. Particular appreciation can also be expressed for the material on which the study was carried out. The data was collected methodically and without bias. The results were consistent and significant and allowed a discussion section full of food for thought. The authors then developed a discussion of the results achieved.

The number and quality of the citations are appropriate; however, the scientific relevance of the article could benefit from an expansion of the same. In the specific advice to add the following quotes:

In the discussion section I would suggest to underlined that the decision to choose a cesarean section rate is associated with the phenomenon of defensive medicine. Thus I suggest adding the following quote: " Fineschi V, Arcangeli M, Di Fazio N, Del Fante Z, Fineschi B, Santoro P, Frati P, Associazione Consulcesi Health And Onlus Futura Ricerca. Defensive Medicine in the Management of Cesarean Delivery: A Survey among Italian Physicians. Healthcare (Basel). 2021 Aug 25;9(9):1097. doi: 10.3390/healthcare9091097. PMID: 34574870; PMCID: PMC8472348.”

English is well structured in syntax and grammar.

Author´s Response: Thank you very much for your comments. We have added the recommended bibliographical reference

Reviewer 3 Report

Thank you so much for doing this very important work. The paper would benefit from a theoretical framework and an editor who is naïve to the topic. There are many sentences that are unclear. I think I understood what you were trying to say, but I am not sure. Especially take care with conjunctions (e.g., despite, however). If you used the established OV instrument, please take time to report on its development and validation.

Abstract:

The conjunctions used in abstract are editorialized (e.g., “but” and “moreover”). I suggest reporting the results more objectively.

“A high percentage of midwives perceived specific clinical practices as OV, the highest degree of perception being associated with certain characteristics of the professional profile.-“ I don’t understand this sentence

Background: a paragraph needs more than 1 sentence.

“Given the high number of women who claim to have experienced OV during child birth”: change the word claim. It implies that it is not true.

“midwives (health professionals who attend the largest number of deliveries)”- in Spain? Requires a citation

This can be better presented in a table: “The following variables were collected. The independent variables were: age, sex, works in a public center (No/Yes), works in a private center (No/Yes), attends home deliveries (No/Yes), works in primary care (No/Yes), annual number births at work center (<1000 births, 1001–3000 births, > 3000 births), if it is a teaching hospital “presence of professionals in training in your center” (No/Yes) , experience as a midwife in birth (<5 years, 106 5–15 years, > 15 years) and previous training in OV (No/Yes).”

Table 2 is very hard to read.

In the aims you talk about measuring prevalence of OV, but when the focus is whether or not people know what it is, it follows that you cannot ask someone who does not know the definition of a construct whether the construct exists. I suggest removing prevalence from the aims and just focusing on midwives perceptions

“The highest number of responses for all 20 practices considered each as very 186 serious; hence, the average score was used to establish the order of severity.” This sentence is unclear as is Table 5. Unclear what order of seriousness is. The word seriousness is unclear. What is the definition that was provided to the participants? It is not a word I usually encounter in English scientific literature.

“Among the possible behaviors that can be considered OV, the majority of the midwives recognized physical aggression as one of them. However, most did not consider 205 that insufficient reporting or lack of introduction to the women could be considered OV, 206 although they did classify these as unacceptable practices. Similarly, delaying care with-207 out justification was not considered OV but inappropriate treatment. Among the clinical 208 practices considered OV, a cesarean section or instrumental birth without clinical justifi-209 cation stands out with the highest score. Regarding professional profile, female midwives, 210 those with less professional experience, those with previous OV training, and those who 211 attend births at home were more likely to perceive specific situations as OV.” The discussion restates the findings without providing contextualization for the findings. You don’t need to restate this given you stated it in results.

This may be due to the essential role that different digital social platforms and communication media are playing, where women have been able to disseminate their experiences and spread awareness of the problem. In addition, the positioning of different professional associations with respect to the subject and the dissemination they have been able to make among their associates has also contributed to the knowledge of the term OV [30–32].—can you also mention here that midwives read professional journals and WHO reports.

Most midwives did not identify OV solely as professional malpractice, as many different groups try to disseminate.—this requires a citation and more information. What do you mean by disseminate? What are the groups?

Our results show that 44% of midwives believe it is necessary to address the problem, although they consider it not being done correctly. Legislation in this regard would help improve clinical practice. Nonetheless, the midwives proposed solutions aimed at greater training, information, and awareness, in line with what is proposed in different studies carried out in Brazil [36–39], Ethiopia [25,26] and an integrative review of the literature carried out by Barbosa Jardim and Modena [40].—these sentences are confusing. Are you stating that you think your participants are incorrect about training and awareness. Maybe it is the word “nonetheless” that is confusing. Your assertion about legislation is not supported. Is that your opinion? If so, please provide some support.

Two paragraphs start with “in line”

In line with what was found by different authors [18,21,22], most of the midwives (92%) reported situations of OV in their workplace. This prevalence is 12 percentage points above the highest prevalence detected in literature, which Costa Cardoso et al. found in a study carried out in Brazil, where 80% of the professionals had witnessed OV behaviors carried out by their peers. However, in the same study, 70% of the professionals denied having carried out OV [22], instead report having treated women with dignity and respect—a professional perception similar to that detected by A Afulani et al. [20] This may be because various clinical practices that can be considered OV are standardized as typical of clinical practice (1,24,41). Hence, the need for training as an element on which to base the prevention and elimination of OV, which is identified in our results and other studies [25,38,40].-- See my previous comment about detecting prevalence in the same study in which you’re determining perception. If you are using an instrument that has been validated, that needs to be described, as well as the instruments used by the authors in the other studies.

Of note, the type of hospital where the midwives worked was not associated with the perception of the existence of OV, in line with what Moyer et al. [46] found.—this is important and deserves more attention.

The discussion needs to be tightened up. I suggest using headings. 2 paragraphs on training. 1-2 on profile of those who recognize OV. Work on citing the literature. At this point, the writing is unclear and confusing.

Author Response

RESPONSE TO REVIEWER3

2 March 2023

Dear Editor of International Journal of Environmental Research and Public Health,

Thank you very much for the opportunity to revise and improve the Manuscript ID: ijerph-2175134 entitled “Violence from the perspective of midwives”

First, we want to express our sincere thanks to the reviewers, as well as their constructive comments. Their comments have allowed us not only to significantly improve the manuscript, but also to reflect on future research in the field of in women' health.

Following this letter, we detail point-by-point our comments in response to the suggestions of the reviewers and the changes that we have made in the revised version of our manuscript. In the revised manuscript, we have highlighted the modifications made to the original text. We hope that the improvements made along with the associated responses below, deserve the definitive approval of the Editorial Team International Journal of Environmental Research and Public Health. If not, all authors remain at your disposal to resolve any issue.

We look forward to hearing from you at your earliest convenience.

Sincerely,

Juan Miguel Martínez Galiano

Reviewer 3: Comments and Suggestions for Authors:

(x) Extensive editing of English language and style required

Author´s Response:  Than you very much for your comments. Dr. Ingrid de Ruiter is a Medical Writer and Public Health Researcher with a background in Clinical Medicine and Public Health. Professional Member of European Medical Writers Association and American Medical Writers Association. She has carefully checked the manuscript. We think no more grammatical errors appear.

Thank you so much for doing this very important work. The paper would benefit from a theoretical framework and an editor who is naïve to the topic. There are many sentences that are unclear. I think I understood what you were trying to say, but I am not sure. Especially take care with conjunctions (e.g., despite, however). If you used the established OV instrument, please take time to report on its development and validation.

Author´s Response: Thank you very much for your comments. We have addressed all your suggestions and recommendations that we believe have improved the manuscript.

Abstract:

The conjunctions used in abstract are editorialized (e.g., “but” and “moreover”). I suggest reporting the results more objectively.

“A high percentage of midwives perceived specific clinical practices as OV, the highest degree of perception being associated with certain characteristics of the professional profile.-“ I don’t understand this sentence

Author´s Response: Thank you very much for your comments. We have rewritten it to make it clearer.

Background: a paragraph needs more than 1 sentence.

“Given the high number of women who claim to have experienced OV during child birth”: change the word claim. It implies that it is not true.

“midwives (health professionals who attend the largest number of deliveries)”- in Spain? Requires a citation

Author´s Response: We have rewritten this, and included a reference to support the latter statement.

This can be better presented in a table: “The following variables were collected. The independent variables were: age, sex, works in a public center (No/Yes), works in a private center (No/Yes), attends home deliveries (No/Yes), works in primary care (No/Yes), annual number births at work center (<1000 births, 1001–3000 births, > 3000 births), if it is a teaching hospital “presence of professionals in training in your center” (No/Yes) , experience as a midwife in birth (<5 years, 106 5–15 years, > 15 years) and previous training in OV (No/Yes).”

Table 2 is very hard to read.

Author´s Response: We have described the operationalization of the variables according to the guidelines of the journal. We have rewritten this.

In the aims you talk about measuring prevalence of OV, but when the focus is whether or not people know what it is, it follows that you cannot ask someone who does not know the definition of a construct whether the construct exists. I suggest removing prevalence from the aims and just focusing on midwives perceptions

Author´s Response: Thank you very much for your comments. We have removed it

“The highest number of responses for all 20 practices considered each as very 186 serious; hence, the average score was used to establish the order of severity.” This sentence is unclear as is Table 5. Unclear what order of seriousness is. The word seriousness is unclear. What is the definition that was provided to the participants? It is not a word I usually encounter in English scientific literature.

Author´s Response: We have changed the adjective serious to grave. The order of seriousness ranks the clinical practices according to the perceived severity of OV. This has been clarified in the text and in Table 5.

“Among the possible behaviors that can be considered OV, the majority of the midwives recognized physical aggression as one of them. However, most did not consider 205 that insufficient reporting or lack of introduction to the women could be considered OV, 206 although they did classify these as unacceptable practices. Similarly, delaying care with-207 out justification was not considered OV but inappropriate treatment. Among the clinical 208 practices considered OV, a cesarean section or instrumental birth without clinical justifi-209 cation stands out with the highest score. Regarding professional profile, female midwives, 210 those with less professional experience, those with previous OV training, and those who 211 attend births at home were more likely to perceive specific situations as OV.” The discussion restates the findings without providing contextualization for the findings. You don’t need to restate this given you stated it in results.

Author´s Response: We have rewritten this. It is a synthesis of the results that in the structure of an article begins the discussion section.

This may be due to the essential role that different digital social platforms and communication media are playing, where women have been able to disseminate their experiences and spread awareness of the problem. In addition, the positioning of different professional associations with respect to the subject and the dissemination they have been able to make among their associates has also contributed to the knowledge of the term OV [30–32].—can you also mention here that midwives read professional journals and WHO reports.

Author´s Response: We have rewritten this sentence to show that midwives obtain information directly from professional journals and WHO reports.

Most midwives did not identify OV solely as professional malpractice, as many different groups try to disseminate.—this requires a citation and more information. What do you mean by disseminate? What are the groups?

Author´s Response: We have rewritten this for clarity.

Our results show that 44% of midwives believe it is necessary to address the problem, although they consider it not being done correctly. Legislation in this regard would help improve clinical practice. Nonetheless, the midwives proposed solutions aimed at greater training, information, and awareness, in line with what is proposed in different studies carried out in Brazil [36–39], Ethiopia [25,26] and an integrative review of the literature carried out by Barbosa Jardim and Modena [40].—these sentences are confusing. Are you stating that you think your participants are incorrect about training and awareness. Maybe it is the word “nonetheless” that is confusing. Your assertion about legislation is not supported. Is that your opinion? If so, please provide some support.

Author´s Response: We have rewritten this for clarity and removed “nonetheless”.

Two paragraphs start with “in line”

In line with what was found by different authors [18,21,22], most of the midwives (92%) reported situations of OV in their workplace. This prevalence is 12 percentage points above the highest prevalence detected in literature, which Costa Cardoso et al. found in a study carried out in Brazil, where 80% of the professionals had witnessed OV behaviors carried out by their peers. However, in the same study, 70% of the professionals denied having carried out OV [22], instead report having treated women with dignity and respect—a professional perception similar to that detected by A Afulani et al. [20] This may be because various clinical practices that can be considered OV are standardized as typical of clinical practice (1,24,41). Hence, the need for training as an element on which to base the prevention and elimination of OV, which is identified in our results and other studies [25,38,40].-- See my previous comment about detecting prevalence in the same study in which you’re determining perception. If you are using an instrument that has been validated, that needs to be described, as well as the instruments used by the authors in the other studies.

Of note, the type of hospital where the midwives worked was not associated with the perception of the existence of OV, in line with what Moyer et al. [46] found.—this is important and deserves more attention.

Author´s Response: We have added a further comment to this point.

The discussion needs to be tightened up. I suggest using headings. 2 paragraphs on training. 1-2 on profile of those who recognize OV. Work on citing the literature. At this point, the writing is unclear and confusing

Author´s Response: Thank you very much for your comments. We have prepared the article according to the guidelines of the journal.

Round 2

Reviewer 2 Report

After examining the scientific study, the following considerations may be made. The scientific study is well structured in all its parts. In particular, the premises with which the authors introduced the analysis are clear. Equally clear are the objectives that led the authors to carry out this study and the section on materials and methods. Particular appreciation can also be expressed for the material on which the study was carried out. The data was collected methodically and without bias. The results were consistent and significant and allowed a discussion section full of food for thought. The authors then developed a discussion of the results achieved.After examining the scientific study, the following considerations may be made. The scientific study is well structured in all its parts. In particular, the premises with which the authors introduced the analysis are clear. Equally clear are the objectives that led the authors to carry out this study and the section on materials and methods. Particular appreciation can also be expressed for the material on which the study was carried out. The data was collected methodically and without bias. The results were consistent and significant and allowed a discussion section full of food for thought. The authors then developed a discussion of the results achieved.